



# Process-oriented modelling to identify main drivers of erosion-induced carbon fluxes

Florian Wilken[1,2,3], Michael Sommer[3,4], Kristof Van Oost[5], Oliver Bens[6], and Peter Fiener[1]

[1]Institute for Geography, Universität Augsburg, Augsburg, Germany
[2]Chair of Soil Protection and Recultivation, Brandenburg University of Technology Cottbus-Senftenberg, Germany
[3]Institute of Soil Landscape Research, Leibniz-Centre for Agricultural Landscape Research ZALF e.V., Müncheberg, Germany
[4]University of Potsdam, Institute of Earth and Environmental Sciences, Potsdam, Germany
[5]Earth & Life Institute, TECLIM, Université catholique de Louvain, Louvain-la-Neuve, Belgium
[6]German Research Centre for Geosciences GFZ, Helmholtz Centre Potsdam, Germany

*Correspondence to*: Peter Fiener (fiener@geo.uni-augsburg.de)

**Abstract.** Over the last few decades, modelling soil erosion and carbon redistribution has received great attention due to large uncertainties and conflicting results. For a process-oriented representation of event dynamics, coupled soil-carbon erosion models have been developed. However, there are currently few models that represent tillage erosion, preferential water erosion and transport of different carbon fractions (e.g. mineral bound carbon, carbon encapsulated by soil aggregates). We couple a

process-oriented multi-class sediment transport model with a carbon turnover model (MCST-C) to identify relevant redistribution processes for carbon dynamics. The model is applied for two arable catchments (3.7 and 7.8 ha) located in the Tertiary hills about 40 km north of Munich, Germany. Our findings indicate the following: (i) Redistribution by tillage has a large effect on erosion-induced vertical carbon fluxes and has a large carbon sequestration potential. (ii) Water erosion has a minor effect on vertical fluxes, but episodic SOC delivery controls the long-term erosion-induced carbon balance. (iii)

Delivered sediments are highly enriched in SOC compared to their parent material, and sediment delivery is driven by event size and catchment connectivity. (iv) Soil aggregation enhances SOC deposition due to the transformation of highly mobile carbon rich fine primary particles into rather immobile soil aggregates.



## 1 Introduction

Soil organic carbon (SOC) is the largest terrestrial carbon (C) pool and has been identified as a cornerstone for the global C cycle. Globally, approx. 1400 Pg C is stored in the upper meter of soil, with approx. 700 Pg C in the upper 0.3 m (Hiederer and Köchy, 2011). As a result, exchange rates between soil and the atmosphere are a major concern with regards to climate change (Polyakov and Lal, 2004a). Earth system model based estimates for terrestrial C storage in the year 2100 vary widely, ranging from a sink of approx. 8 Pg C yr$^{-1}$ to a source of approx. 6 Pg C yr$^{-1}$ (Friedlingstein et al., 2014). This large uncertainty might even increase if process-levels that are to this point not yet implemented to current models are taken into account (Doetterl et al., 2016). One such process is the lateral redistribution of SOC via erosion processes and the effect this has on vertical C fluxes. Global estimates of erosion-induced C fluxes show conflicting results, ranging from a source of 1 Pg C yr$^{-1}$ to a sink of the same magnitude (for recent reviews see Doetterl et al., 2016; Kirkels et al., 2014). The main reasons for these large differences are a lack of appropriate data (Prechtel et al., 2009), oversimplified modelling approaches that ignore important processes, and differences in measuring approaches, e.g. extrapolating from arable plots (Hooke, 2000; Myers, 1993; Pimentel et al., 1995) vs measuring continental delivery in river systems (Berhe et al., 2007; Wilkinson and McElroy, 2007). Most challenging in developing and especially testing models that couple process-oriented SOC redistribution with SOC dynamics are the different spatial and temporal scales of the processes at play (Doetterl et al., 2016). Process-oriented erosion models need event-based data to be validated, while SOC dynamics can hardly be observed on time scales smaller than several decades. Consequently, most existing models that couple soil erosion and SOC turnover processes are based on long-term, USLE-type erosion models that ignore event dynamics. The most widespread of these is SPEROS-C, which was applied on scales ranging from micro- to mesoscale catchments (Fiener et al., 2015; Nadeu et al., 2015; Van Oost et al., 2005b).

The conventional approach to modelling coupled soil erosion and SOC turnover is to treat SOC as a stable part of the bulk parent soil and statistically model (long-term) erosion. However, this approach is likely to lead to biased estimates of both water erosion-induced SOC redistribution and its effect on vertical C fluxes. Numerous studies have shown that the transport of SOC is selective (Schiettecatte et al., 2008), controlled by event characteristics (Sharpley, 1985; Van Hemelryck et al., 2010) and soil aggregation (Hu and Kuhn, 2014; 2016). The enrichment of SOC during transport has been explicitly addressed by a few modelling studies, using different approaches (Fiener et al., 2015; Lacoste et al., 2015). The effects of tillage erosion on vertical C fluxes have not yet been evaluated in detail, although a representation has been accounted for in some modelling studies (Lacoste et al., 2015; Van Oost et al., 2005a).

The aim of this study is to couple a spatially distributed, process-oriented and event-based water erosion model with a tillage erosion model and a SOC turnover model in order to analyse the importance of individual erosion processes upon the erosion-induced C balance of agricultural catchments. The study intends to identify relevant processes that should be implemented in less data demanding, more parsimonious models.



## 2 Materials and Methods

### 2.1 Test site

The test site is located about 40 km north of Munich in the Tertiary hills, an intensively used agricultural area in Southern Germany. The site consists of two small arable catchments (48°29´ N, 11°26´ E; Fig. 1), catchment C1 and C2, covering an

area of 3.7 and 7.8 ha, respectively. The rolling topography ranges from 454 to 496 m above sea level with a mean slope of 4.2° (±0.6°) for catchment C1 and 5.3° (±1.7°) for catchment C2. The soil landscape is characterized by Cambisols and Luvisols (partly redoximorphic), both developed from Loess. Furthermore, Colluvic Regosols have developed in depressional areas due to long-term soil translocation processes. In both catchments, the dominant topsoil textures are loam and silty loam with a median grain size diameter between 12.5 and 16.0 µm (Sinowski and Auerswald, 1999). The average SOC content of

the Ap horizons is 3.7 kg $m^{-2}$. The mean annual temperature and precipitation is 8.4°C and 834 mm, respectively (measured 1994 to 2001). Agricultural management at the research farm is dedicated to soil conservation: the main cropping principle is to keep soil covered by vegetation or residues as long as possible. The crop rotation during the project was winter wheat (*Triticum aestivum* L.) – maize (*Zea mays* L.) – winter wheat – potato (*Solanum tuberosum* L.). This crop rotation allowed for the cultivation of mustard (*Sinapis alba* L.) cover crops before each row crop (i.e. potato and maize). For implementation,

potato ridges were formed before mustard sowing and, later, potato was directly sown into the ridges covered by winterkilled mustard. Maize, on the other hand, was directly sown into the winterkilled mulched mustard (Auerswald et al., 2000). For the established mulch tillage system, the main soil tillage operation was performed with a chisel cultivator (tillage depths approx. 0.2 m). To avoid soil compaction and depressions, which could potentially induce concentrated runoff, wide and low pressure tires were used on all farming machines (e.g. Fiener and Auerswald, 2007b). Catchment C1 drains a large field with an approx.

5 m wide grass filter strip along its downslope border, whereas catchment C2 consists of two fields draining into an approx. 300 m long and 30-40 m wide grassed waterway (Fig. 1).

### 2.2 Model description

For our study, we coupled three different models: (i) the process-oriented Multi-Class Sediment Transport Model (Fiener et al., 2008; Van Oost et al., 2004; Wilken et al., 2016), a spatially distributed and event-based water erosion model with a specific

emphasis on grain size selectivity using the Hairsine and Rose equations (Hairsine et al., 1992; Hairsine and Rose, 1991), (ii) a tillage erosion model following a diffusion-type equation adopted from Govers et al. (1994), which derives tillage erosion from topography and tool-specific tillage erosion coefficients, and (iii) the Introductory Carbon Balance Model (Andrén and Kätterer, 1997; Kätterer and Andrén, 2001), which models SOC turnover. The ICBM calculates yearly SOC dynamics using two SOC pools ("young" and "old") and four C fluxes (C input from plants, mineralisation from the young and the old pool,

and humification). Both the tillage erosion and ICBM model were adapted from SPEROS-C, which couples annual water erosion (based on the RUSLE; Renard et al., 1996), tillage erosion and SOC turnover (Fiener et al., 2015; Nadeu et al., 2015; Van Oost et al., 2005b). In the following, we describe only those features of the coupled MCST-C model (Multi-Class Sediment





Transport and Carbon dynamics model) that had to be adapted in order to couple the models or for the introduction of SOC-specific transport mechanisms. An overview of the main model concepts of MCST-C are given in Fig. 2. For more details regarding the three coupled models and processes modelled therein, we refer to the original publications (see above).

**2.3 Representation of grain size specific soil and associated SOC**

The representation of soil texture and SOC in the model is three dimensional. The horizontal distribution of grain size specific soil and SOC is grid based, while the vertical distribution is represented by ten 10 cm layers. The two uppermost layers are assumed to be homogeneously mixed due to tillage. The grain size distribution is represented in 14 primary particle classes, described by class median particle diameter, particle density and the class proportion relative to the bulk soil (kg/kg). The median class diameter is calculated by a logarithmic function that takes grain diameter class boundaries into account (Scheinost

et al., 1997). The standard procedure (e.g. sieve-pipette method; Casagrande, 1934; DIN, 2002) to determine grain size distributions destroys soil aggregates in a pre-processing step and therefore only represents the primary particle distribution. However, soil aggregation has a large effect on the fall velocity distribution of soils and reduces the transport distance of SOC rich material (Hu and Kuhn, 2014; 2016). Therefore, to account for soil aggregation, two water stable aggregate classes have been introduced following the hierarchy model of Oades (1984), which describes microaggregate formation inside

macroaggregates: silt-sized small microaggregates (6.3 – 53 µm, median diameter D50: 18 µm; Tisdall and Oades, 1982) and microaggregates (53 – 250 µm, D50: 115 µm; Six et al., 2002). In model parametrization, the small microaggregates are exclusively formed out of primary particles with diameters less than 6.3 µm, whereas microaggregates are formed from those with diameters less than 53 µm (i.e. the lower diameter boundary of the aggregate class). As a result, aggregation causes a certain number of primary particles to be moved into the aggregate classes. Hence, the absolute amount of soil aggregation is

controlled by the availability of fine primary particles, i.e. sandy soils are less aggregated compared to clayey soils. Macroaggregates (250-2000 µm) are neglected since they are rather immobile during selective interrill erosion and are assumed to break into smaller aggregates during extreme events with high precipitation kinetic energies (Legout et al., 2005; Oades and Waters, 1991; Tisdall and Oades, 1982). Furthermore, particulate organic matter (POM) is not treated as an individual class, as POM is assumed to be predominantly encapsulated within soil aggregates (Beuselinck et al., 2000; Wang et al., 2013;

Wilken et al., 2016).

SOC transport is associated with various primary particle and aggregate classes. Based on literature (Doetterl et al., 2012; Von Lützow et al., 2007), it is assumed that mineral bound SOC is primarily attached to fine particles (< 6.3 µm) or included in soil aggregates. To keep the mass balance, SOC in water stable aggregates is allocated based on the SOC content of the primary particles that form these aggregates. This leads to a conservative estimate of SOC in aggregates, as measurements show that

aggregates tend to encapsulate more C than found attached to mineral primary particles (Doetterl et al., 2012). As small microaggregates in the model consist solely of primary particles with diameters less than 6.3 µm, their C content equals that of the fine primary particles. Microaggregates show a somewhat smaller C content, since the larger primary particles from which they are also made tend to have less associated SOC.



## 2.4 Continuous tracking of catchment dynamics

In its original version, the MCST model treats events individually without considering changes caused by previous events. For a continuous application, the water erosion module of MCST-C simulates single events and keeps track of the following redistribution related changes in the catchment: spatial and vertical changes in (i) the grain size distribution and (ii) SOC content, and (iii) the development of a rill network, which remains until the next tillage operation. A layer-specific mixture module continuously updates for spatial changes in the vertical grain size distribution and its associated SOC content, changes which are caused by selective redistribution of water and non-selective tillage erosion. In the case of net deposition, new material with a different grain size distribution is added to the top of the plow horizon (layer 1 and 2). Subsequently, the grain size distribution of the plow layer is mixed and assumed to be homogenous. Furthermore, deposition leads to an upward movement of the layer borders such that soil material from the plow layer becomes incorporated into the subsoil layers. Any C content moving below 1 m depth is summarized and assumed to be stable with time. In contrast, erosion lifts new material from the subsoil horizons upwards. Assuming that the deepest horizon represents the original loess, the properties of uplifted soil remain constant, delivering infinite material of the same grain size distribution and C content.

## 2.5 Model validation

For a truly rigid validation of MCST-C, there are numerous long-term data requirements: event-based data for surface runoff, sediment delivery and SOC delivery, long-term data regarding changes in spatially distributed SOC stocks, spatially distributed C loss and gain due to crop harvesting, and C input via plants and manure application. In addition to these validation data requirements, model input data would also be required over decades for a long-term validation. The research farm used in this study has a very comprehensive database available. However, continuous monitoring was 'only' carried out for 8 years (1994 to 2001), and SOC inventories span roughly a decade (first inventory in 1990/91, second in 2001). Therefore, measured changes in SOC stocks are too small to be used for a long-term model validation (requires approx. 50 years; see implementation).

In consequence, we only use the measured continuous event-based surface runoff and sediment delivery from catchment C1 to validate the modelled erosion. The runoff was collected at the lowest point of the catchment (Fig. 1), which was bordered by a small earthen dam. From the dam, the runoff was transmitted via an underground tile outlet (diameter 0.29 m) to a measuring system consisting of a Coshocton-type wheel runoff sampler (for details regarding the procedure and the precision of the measurements see Fiener and Auerswald, 2003). Corresponding precipitation was measured using a tipping bucket rain gauge of 0.2 mm volume resolution. To determine single erosion events, the precipitation data is filtered in two steps: First, all events with cumulative precipitation > 5 mm and without a 6 h pause in recorded precipitation are considered single erosion events. Second, we included all the largest events accounting for 90% of total observed runoff. The model is not able to predict erosion under soil frost; hence, winter events, indicated by air temperatures below zero, are removed.





As the original MCST model was previously tested in catchment C1 (Fiener et al., 2008), we did not explicitly calibrate the surface runoff and erosion model. Instead, observed runoff and sediment delivery data was used to test if our changes to the model still result in a reasonable model performance.

## 2.6 Model implementation

To run and test MCST-C, a variety of measured input data and parameters are required. This input data is partly calculated from measured data at the research farm and partly taken from literature (Table 1; Fig. 2). To model surface runoff and erosion, the most important input data requirements are (i) precipitation, measured at two meteorological stations about 100 to 300 m from the catchments using 0.2 mm tipping buckets, (ii) a LIDAR 5 x 5 m² digital elevation model, (iii) interpolated (5 x 5 m²) soil data taken from a 50 x 50 m raster sampled during the soil survey in 1990/91, and (iv) soil cover data, measured biweekly

during the vegetation period, monthly in autumn and spring, and before and after each soil management operation (1993-1997). A tillage transport coefficient ($k_{til}$) of 169 kg m per pass was utilized for contour tillage by a chisel, following Van Muysen et al. (2000). For SOC redistribution and modelling of vertical C fluxes, the most important model inputs were yields and manure application, a topsoil SOC map (12.5 x 12.5 m²) and assumptions regarding the allocation of C to different texture classes and in different aggregates. As texture and aggregate C allocation was not measured, we took measured data from

Doetterl et al. (2012) and scaled these measurements according to the available bulk SOC (see Section 2.3: Representation of grain size specific soil and SOC). The parameters for the C turnover model are taken from Dlugoß et al. (2012), who worked under similar environmental conditions with loess-burden soils in a small catchment in western Germany. The C turnover decline with depth was determined by an inverse modelling approach and found a mean residence time of approx. 10 years for the young pool and 1360 years for the old pool in the deepest layer (1 m). Further details regarding the monitoring data are

given in Fiener and Auerswald (2003, 2007b) and Fiener et al. (2008).

As indicated above, it is difficult, if not impossible, to identify erosion-induced changes in SOC and vertical C fluxes if measurements or modelling efforts do not cover decadal time spans. Therefore, a 50 year synthetic input data set and parameter set was created for MCST-C in order to analyse C dynamics. This data set is based on the 8 years of measured data used to validate the erosion component of the model. First, a time series of precipitation was established by randomly choosing the

data of one of the eight measured years (see Section 2.5: Model validation) and applying it for the first 42 years of the time series. This was followed by the original 8 measured years to reach the total of 50 years. Next, this precipitation time series was combined with synthetic land use and soil management data representing two full crop rotations (1994 to 2001), which were repeatedly used for all 50 years. This combination leads to a wide variety of precipitation events (time step 1 min) occurring for different daily soil covers by vegetation as major driver of soil erosion. In contrast to the erosion dynamics, C

inputs via plants and manure are repeated every 8 years, which ignores any potential change in management and yields within the modelling period. The synthetic input data were applied for both catchments for the purpose of comparability.



## 2.7 Analysis of process-specific, erosion-induced C fluxes

Various model setups were chosen (Tab. 2) to analyse the effects of different erosion processes upon lateral SOC redistribution and the resulting vertical C fluxes. All of these model runs were compared to the 50 year reference run that was validated for the 8 years monitoring phase at the research farm (1994-2001). In general, we tested the effect of a number of water erosion

processes and compared the relevance of water vs. tillage erosion. Firstly, the critical shear stress of rill initiation ($\tau_{crit}$) was varied by ±50% in comparison to its reference run value (0.9 Pa) in order to change the proportion of interrill vs. rill erosion, where interrill erosion is a selective SOC transport process and rill erosion is unselective. The reference run value for $\tau_{crit}$ was derived from flume experiments in loamy, loess-burden soils (Giménez and Govers, 2002) similar to those found at the test site. Next, the aggregation level was varied in an analogous way to modify the allocation of soil primary particles into the

small microaggregate and microaggregate classes (Fig. 3). In another model run, grain size selectivity was switched off in order to produce a similar behaviour to more parsimonious models, which only erode bulk soil (Tab 2). To analyse the sensitivity of C fluxes to water and tillage erosion, we first compared model runs with pure water or pure tillage erosion. Secondly, we varied the reference run $k_{til}$ coefficient of 169 kg m per pass by ±50%. All model runs altered only a single parameter, with all other parameters retaining their reference run values. Parameter variations and the abbreviations for each

of the model runs are given in table 2.

## 2.8 Analysis of erosion-induced C fluxes

To compare vertical C fluxes from erosional and depositional sites, the corresponding total and mean C flux was calculated on an annual basis. To isolate the C fluxes that result solely from erosion processes, we first calculate all vertical C fluxes excluding erosion processes and then subtract these from the vertical C fluxes including erosion processes. In the following

results section, positive C fluxes indicate an erosion-induced C gain for the catchment (input to the soil), while negative fluxes indicate an erosion-induced loss (from soil to the atmosphere or SOC delivery from the catchment by runoff). Subsequently, erosional and depositional sites were spatially subdivided and an average vertical C flux in kg C m$^{-2}$ was calculated. Finally, the erosion-induced C balance of the catchment was calculated as the sum of the total vertical C flux and laterally delivered SOC.

## 3 Results

### 3.1 Validation

A number of goodness-of-fit parameters (Table 3) indicate a sufficient model performance to simulate event runoff and sediment delivery for the 8 year observation period. The Nash-Sutcliffe efficiency and coefficient of determination for runoff (NSE = 0.83; $R^2$ = 0.94) and sediment delivery (NSE = 0.92; $R^2$ = 0.95) are particularly satisfactory. However, a root mean

square error of 165 kg/ha for sediment delivery indicates difficulties in predicting some small events.



## 3.2 Long-term erosion induced C fluxes

The simulated tillage and water erosion shows distinct spatial patterns (Fig. 4). The highest rates of tillage erosion are found along the upslope boundaries of the arable field and on hill tops. The main areas for tillage induced deposition are downslope arable field boundaries and in concavities (Fig. 4). Due to the well-established soil conservation system, water erosion takes place over a much smaller spatial extent and is limited to the main hydrological flow path, while deposition is dominantly found in the vegetated filter strip and grassed waterway (Fig. 4).

The reference run (validated against sediment delivery in catchment C1, 1994-2001) shows positive vertical C fluxes at erosional sites over the 50 year simulation period, with a cumulative flux of 40 g m$^{-2}$ (50 yr)$^{-1}$ in C1 and to 59 g m$^{-2}$ (50 yr)$^{-1}$ in C2 (Fig. 5: Ero1, Ero2). The depositional C fluxes show a cumulative C loss of 27 g m$^{-2}$ (50 yr)$^{-1}$ and 30 g m$^{-2}$ (50 yr)$^{-1}$ for C1 and C2, respectively (Fig. 5: Dpo1, Dpo2). Lateral SOC delivery is driven by three heavy erosion events causing 58% and 53% of the total SOC delivery in C1 and C2, respectively. The total SOC delivery in C1 is 15.6 g m$^{-2}$ (50 yr)$^{-1}$ and in C2 is -6.5 g m$^{-2}$ (50 yr)$^{-1}$ (Fig. 5: Del1, Del2). In C1, the source function of lateral SOC delivery exceeds the sink function of vertical SOC sequestration and leads to a net C source of 5.7 g m$^{-2}$ (50 yr)$^{-1}$ (Fig. 5: Bal1, Bal2). In contrast, catchment 2 is a net C sink of 4.6 g m$^{-2}$ (50 yr)$^{-1}$.

The event based SOC enrichment in delivered sediments, compared to parent soil, ranges from 1.1 to 2.7 (2.4 mean) for C1 and from 2.5 to 2.7 (2.7 mean) for C2 over the 50 year time span (Fig. 6). Subdividing the events into tertiles (33%-parts) according to sediment delivery, the mean enrichment in C1 is 2.5 (n=67) for the low tertile (i.e. smallest 33% of all event-specific sediment delivery masses), 1.4 (n=6) for the mid tertile and 1.2 (n=2) for the high tertile (Fig. 6). In contrast, more or less no variation in SOC enrichment was modelled for C2 (Fig. 6).

## 3.3 Importance of individual erosion processes for long-term erosion-induced C fluxes

Vertical C fluxes show a large response to changes in the $k_{til}$ coefficient but a negligible response to varying levels of water erosion (Fig. 5: Ero1, Ero2, Dpo1, Dpo2). Cumulative C flux at erosional and depositional sites is found to be lowest when no tillage (Til$_{off}$) is simulated and highest for strong tillage (Til$_{hi}$). When pure tillage erosion is simulated (Wa$_{off}$) in catchment C1, a C sequestration of 7 g m$^{-2}$ (50 yr)$^{-1}$ is simualted (Fig. 5: Bal1). The majority of processes in catchment C2 lead to an erosion-induced C gain for the catchment. The highest C sequestration in catchment C2 is found for high tillage erosion (Til$_{hi}$: 10.3 g m$^{-2}$ (50 yr)$^{-1}$). In contrast, catchment C2 acts as a source when there is no tillage (Wa$_{off}$: -4.8 g m$^{-2}$ (50 yr)$^{-1}$), as well as when tillage erosion is low (Til$_{lo}$: -0.4 g m$^{-2}$ (50 yr)$^{-1}$; Fig. 5: Bal2).

Lateral SOC delivery is solely caused by water erosion. The model shows its smallest levels of lateral SOC delivery when grain size selectivity is ignored (Gs$_{off}$) and delivered sediments therefore have the same SOC concentration as the parent material (C1: -10 g m$^{-2}$ (50 yr)$^{-1}$; C2: -2.4 g m$^{-2}$ (50 yr)$^{-1}$). This effect is less pronounced for catchment C2 (Fig. 5: Del1, Del2). Catchment C1 shows the largest SOC delivery when the threshold for rill initiation is low (Ril$_{lo}$: -26.3 g m$^{-2}$ (50 yr)$^{-1}$). In catchment C2, the highest lateral SOC delivery is achieved when there is assumed to be no soil aggregation (Agg$_{off}$: -13.0 g





m$^{-2}$ (50 yr)$^{-1}$). If water erosion is taken into account, catchment C1 is a net C source ranging from 1.3 (Gs$_{off}$) to 14.2 (Ril$_{hi}$) g m$^{-2}$ (50 yr)$^{-1}$. In contrast, the tillage induced sequestration potential of catchment C2 exceeds SOC delivery in most water erosion model runs, leading to a positive erosion-induced C balance (sink) as long as soil aggregation is included (Agg$_{off}$: -1 g m² (50 yr)$^{-1}$; Fig. 5: Bal1-Bal2).

Variations in SOC enrichment of delivered sediments is generally rather small for all model runs (Fig. 6). The most pronounced effect on SOC enrichment results from different aggregation levels (Agg$_{off}$, Agg$_{lo}$, Agg$_{hi}$). However, differences in SOC enrichment were much more pronounced between the catchments. While C2 show high enrichment ratios (> 2.5) for all events, the enrichment ratios strongly decline with increasing event size in C1 (Fig. 6: B-C).

## 4 Discussion

### 4.1 Vertical C fluxes

Tillage erosion dominates the erosion-induced vertical C fluxes in both catchments. Without water erosion (Wa$_{off}$), total tillage erosion-induced C sequestration potential was rather low, at 7 and 9 g m² (50 yr)$^{-1}$ in catchment C1 and C2, respectively. The higher sequestration potential in catchment C2 results from steeper slopes and more field boundaries, where tillage erosion is most pronounced (Fig. 4). This off-sets its smaller relative proportion of arable land. However, this field boundary effect (Fig.
4) might be overestimated as we did not update the digital elevation model during the 50 year simulation period. The response of vertical C fluxes to changes in tillage erosion strength (Til$_{lo}$; Til$_{hi}$) further underlines the dominance of tillage redistribution in determining these fluxes (Fig. 5). This dominance results, in part, from the soil conservation system established at the research farm. Indeed, when compared to conventional soil management practices, water erosion was reduced by roughly a factor of 20 (Fiener and Auerswald, 2007a) in both catchments and the k$_{til}$ coefficient was about 3-times smaller (Van Oost et
al., 2006) as a result of the soil conservation system. However, independent from the soil tillage management, it is obvious that tillage erosion needs to be taken into account for reasonable estimates of vertical erosion-induced C fluxes on arable land (see also Van Oost et al., 2005a). Moreover, it should be noted that modelling tillage erosion is associated with large uncertainties since it is controlled by a large number of parameters (e.g. tool geometry and type, up-down or contour tillage, speed, depth, soil characteristics; Van Muysen et al., 2000; Van Oost and Govers, 2006). This uncertainty is illustrated by the
large range of $k_{til}$ coefficients which can be found in literature (e.g. for chisel ktil: 70 to 657; Van Oost and Govers, 2006). Interestingly, different water erosion processes hardly affected the vertical erosion induced C fluxes. This is even true for model parametrisations with very pronounced rill erosion (Ril$_{hi}$) and large sediment fluxes, because rills only affect a small area. Deposition is also restricted to a small number of raster cells (Fig. 4), particularly the grassed waterway of catchment C2. Overall, to achieve accurate estimates of vertical erosion-induced C fluxes, it seems to be more important to improve the
representation of tillage erosion in the model, rather than focusing on detailed process-oriented water erosion modelling, which is less important for vertical C fluxes.



## 4.2 Lateral C fluxes

In contrast to vertical C fluxes, lateral erosion-induced C fluxes are substantially affected by a number of event-specific processes. To assess these processes, a spatially distributed process-oriented modelling approach is needed.

Our synthetic 50 year data set (based on the 1994-2001 observations) produces three large SOC delivery events, representing

nearly 60% of the total SOC delivery in both catchments (Fig. 5: Del1-Del2). This underlines the importance of accounting for individual events, particularly for the enrichment of SOC in delivered sediment (Fig. 6). However, it should be noted that SOC enrichment is mostly affected by catchment characteristics (Fig. 6: B-C). While catchment C1 follows the expected behaviour, i.e. decreasing SOC enrichment with increasing event size (Auerswald and Weigand, 1999; Menzel, 1980; Polyakov and Lal, 2004b; Sharpley, 1985), and is in good agreement with the results of Wang et al. (2010) for similar soils in the Belgian

loam belt, event size had hardly any effect on the SOC enrichment in catchment C2, where any larger particles, including aggregates, are deposited in the grassed waterway due to consistently high hydraulic roughness throughout the year. Hence, a parsimonious approach solely relating annual erosion magnitude to SOC enrichment (e.g. Fiener et al., 2015 using the model SPEROS-C) might fail on the landscape scale due to varying inter-field connectivity characteristics of catchments. Underlining the results of recent studies (e.g. Hu and Kuhn, 2016), it seems to be essential to take detailed processes into account during

erosion, transport and deposition in order to accurately capture the SOC enrichment of delivered sediments. In our modelling example, neglecting enrichment would lead to a 36% underestimation of the total SOC delivery in catchment C1 and an even more extreme 63% underestimation in catchment C2. This large difference between catchment C1 and C2 suggests that the relevance of SOC enrichment in delivered sediments is not only controlled by event size but also by the catchment connectivity to the outlet.

SOC enrichment in delivered sediments is mainly controlled by the physical properties (e.g., soil texture) of the parent material (Foster et al., 1985). Soil aggregation transforms unconsolidated fine primary particles, a highly mobile SOC fraction, into soil aggregates, a fraction in which SOC is far less mobile. Hu and Kuhn (2016) showed that soil aggregation reduces the transport distance and potentially enhances terrestrial SOC deposition up to 64%. We found a similar trend: upon increasing the aggregation level of the model from non-aggregated ($Agg_{off}$) to heavily aggregated ($Agg_{hi}$) soil conditions, we found an

increase in SOC deposition for both catchment C1 (47%) and C2 (83%). However, while soil texture clearly plays an important role, intra-field connectivity can be the dominant process driving lateral SOC delivery on the landscape scale. This is demonstrated by catchment C2, which shows its largest SOC delivery when it is assumed that there is no soil aggregation. Unfortunately, representing soil aggregation in models is challenging due to a pronounced seasonality (Angers and Mehuys, 1988; Coote et al., 1988; Six et al., 2004; Wang et al., 2010) and complex spatial patterns related to soil nutrients, moisture,

grain size distribution, management practices, erosion and soil biota (Denef et al., 2002). Especially for landscape scale applications, this high degree of complexity needs to be substantially reduced in a conceptual way.

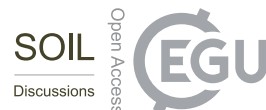

### 4.3 Erosion-induced C balance of different catchments

Under the same precipitation and field conditions, the simulated erosion-induced C balance of catchment C1 and C2 show opposing results (Fig. 5: Bal1-Bal2). While catchment C1 acts as a C source for the majority of simulated processes (controlled primarily by SOC delivery), the presence of the grassed waterway for catchment C2 substantially reduces lateral SOC delivery

and leads the catchment to function as a C sink for most simulated processes. For both catchments, the majority of simulation years show a positive erosion-induced C balance (sink). However, three heavy erosion events in catchment C1 exceeded the positive cumulative vertical flux. Therefore, we underline that any analysis of landscape scale erosion-induced C balances must consider inter-field connectivity.

### 5 Conclusions

In this study, the effect of individual SOC redistribution processes on SOC dynamics is assessed by utilizing a coupled process-oriented erosion and C turnover model. The erosion component of the model was successfully validated against a continuous 8-year data set of surface runoff and sediment delivery. The model was able to estimate the relevance of different processes in terms of their impact on vertical and lateral C fluxes for two catchments with distinct characteristics over an artificial time series of 50 years. We found that tillage erosion dominates on-field soil redistribution and vertical erosion-induced C fluxes

on arable land, while water erosion processes have a much more limited effect. However, episodic lateral SOC delivery is critically important for the carbon balance. Ignoring SOC enrichment in delivered sediments leads to a pronounced underestimation of delivered SOC. Soil aggregates substantially reduce SOC delivery by turning highly mobile fine primary particles into less mobile soil aggregates. In general, the erosion-induced C balance is largely affected by intra-field deposition related to catchment connectivity.

Our results underline the importance of having an accurate and spatially distributed representation of tillage erosion. The episodic nature of water erosion calls for a sufficiently long simulation period and the inclusion of grain size selective transport in order to address the enrichment of delivered SOC. Furthermore, we stress the need for future investigations on seasonal and spatial variations in soil aggregation for a conceptual model implementation.

### Acknowledgements

The study was supported by the Terrestrial Environmental Observatory TERENO-Northeast of the Helmholtz Association. We would like to acknowledge the large number of scientists and technicians who collected the data used in this study, which was funded by the Bundesministerium für Bildung, Wissenschaft, Forschung und Technologie (BMBF No. 0339370) and the Bayerische Staatsministerium für Unterricht und Kultus, Wissenschaft und Kunst.



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



**Tables**

Table 1. **Main input data and parameters used in the Multi-Class Sediment Transport and Carbon dynamics model (MCST-C).**

| Description | Unit | Temporal resolution | Range / value |
|---|---|---|---|
| Digital elevation model | m | static | (5 x 5 m) 454 - 496 |
| Land use | - | day | - |
| Soil cover | % | biweekly | 0 - 100 |
| Curve number per crop to be modified by cover and soil crusting | - | day | 38 - 88 |
| Tillage roughness and direction | m | vegetaion period | 0 - 0.25 |
| Hydraulic roughness arable | s m$^{-1/3}$ | biweekly | 0.016 - 0.101 |
| Hydraulic roughness grass strip | s m$^{-1/3}$ | static | 0.20 |
| Yield | kg m$^{-2}$ | at harvest | 0.6 - 4.3 |
| Manure | kg C m$^{-2}$ | at fertilisation | 0 - 0.13 |
| Tillage operation | - | day | - |
| Soil bulk density | kg/m³ | static | 1350 |
| Initial texture | μm | static | 0.04 - 2000 |
| Primary particle density | kg/m³ | static | 2650 |
| Small microaggragate density | kg/m³ | static | 1300 |
| Microaggragate density | kg/m³ | static | 1300 |
| Small microaggragate median diameter | μm | static | 18 |
| Microaggragate median diameter | μm | static | 115 |



**Table 2. Model parameterisation to analyse the effects of different erosion processes upon C fluxes. Model runs are abbreviated as follows: reference run (*Ref*), without tillage erosion (*Til_off*), water erosion without grain size selectivity (*GS_off*), low threshold for rill initiation (*Ril_lo*), high threshold for rill initiation (*Ril_hi*), without soil aggregation (*Agg_off*), low soil aggregation (*Agg_lo*), high soil aggregation (*Agg_hi*), without water erosion (*Wa_off*), low tillage erosion (*Til_lo*), and high tillage erosion (*Til_hi*).**

| Processes | Parameter [unit] | *Ref* | *Til_off* | *GS_off* | *Ril_lo* | *Ril_hi* | *Agg_off* | *Agg_lo* | *Agg_hi* | *Wa_off* | *Til_lo* | *Til_hi* |
|---|---|---|---|---|---|---|---|---|---|---|---|---|
| ***Water erosion*** | | | | | | | | | | | | |
| with vs. w/o tillage erosion | [-] | +[#] | - | + | + | + | + | + | + | + | + | + |
| with vs. w/o grain size selectivity | [-] | + | + | - | + | + | + | + | + | + | + | + |
| varying rill/interrill erosion | $\tau_{crit}$[‡] [Pa] | 0.9 | 0.9 | 0.9 | 1.35 | 0.45 | 0.9 | 0.9 | 0.9 | 0.9 | 0.9 | 0.9 |
| varying small micro & microaggregates | [%] | 60 | 60 | 60 | 60 | 60 | 0 | 30 | 90 | 60 | 60 | 60 |
| ***Tillage erosion*** | | | | | | | | | | | | |
| with vs. w/o water erosion | [-] | + | + | + | + | + | + | + | + | - | + | + |
| varying tillage intensity | $k_{til}$[*] [kg m per pass] | 169 | 0 | 169 | 169 | 169 | 169 | 169 | 169 | 169 | 85 | 254 |

5   [#] + and - indicates if a process is modelled or not; [‡] critical shear stress for rill initiation; [*] tillage erosion coefficient.



**Table 3: Model performance, as described by the Nash-Sutcliffe efficiency (NSE; Nash and Sutcliffe, 1970), root mean square error (RMSE), coefficient of determination (R²), and Spearmans rank correlation coefficient (RHO).**

|  | NSE | RMSE | R² | RHO |
|---|---|---|---|---|
| Runoff | 0.83 | 56 m³/ha | 0.94 | 0.89 |
| Sediment delivery | 0.92 | 165 kg/ha | 0.95 | 0.71 |



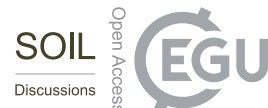

**Figures**

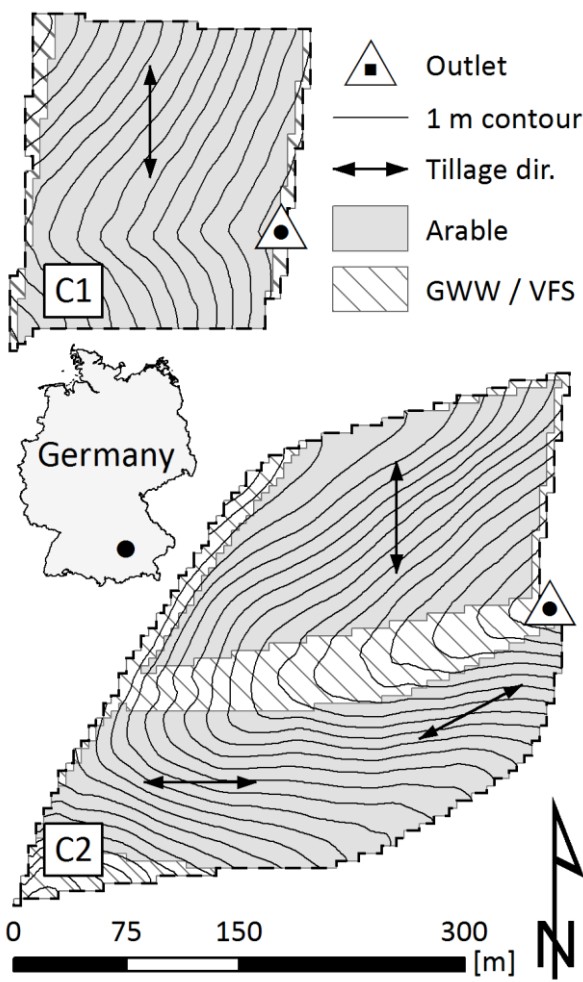

**Figure 1: Land use, topography and tillage direction for modelled catchments C1 and C2. In catchment C2, a grassed waterway (GWW) is located along the thalweg, while vegetated filter strips (VFS) are located along the upslope and downslope field borders.**




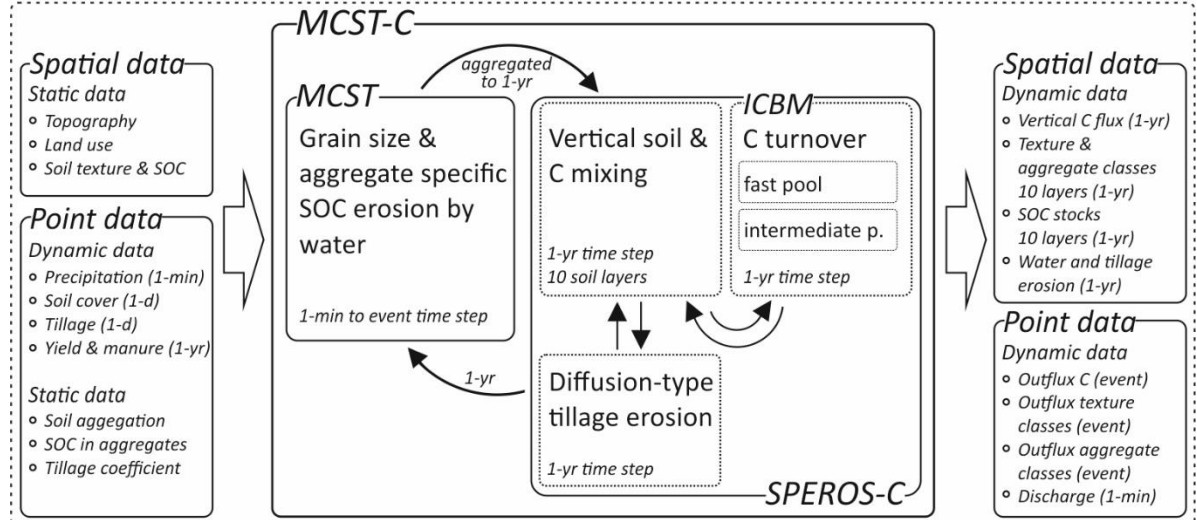

**Figure 2: Modelling scheme of the Multi-Class Sediment Transport and Carbon dynamics model (MCST-C)**




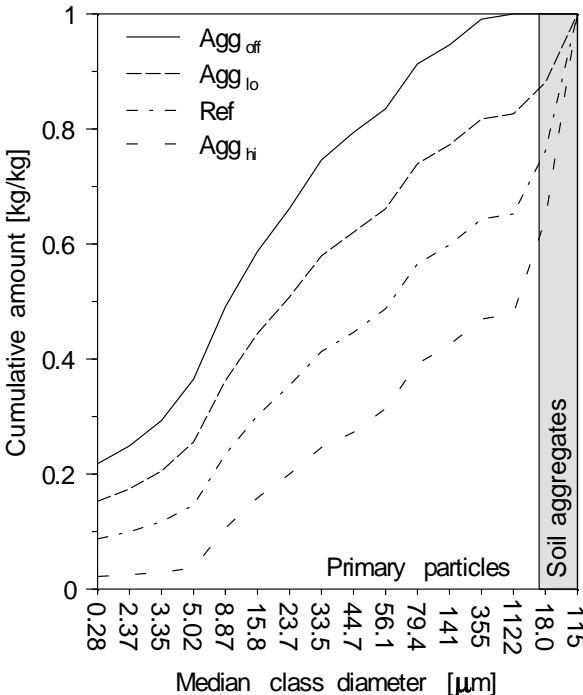

**Figure 3: Median class diameter distribution in the plow layer assuming different aggregation levels, as described in Table 2.**




**Figure 4: Spatial patterns of tillage and water erosion for the 50 year simulation period of the reference run.**



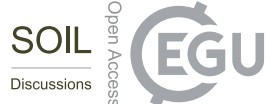

**Figure 5: Simulated cumulative vertical C fluxes for erosional (Ero1, Ero2) and depositional (Dpo1, Dpo2) sites; lateral C delivery (Del1, Del2) and overall catchment C balance (Bal1, Bal2) for catchment C1 and C2. For details regarding the model runs and corresponding abbreviations see Table 2.**





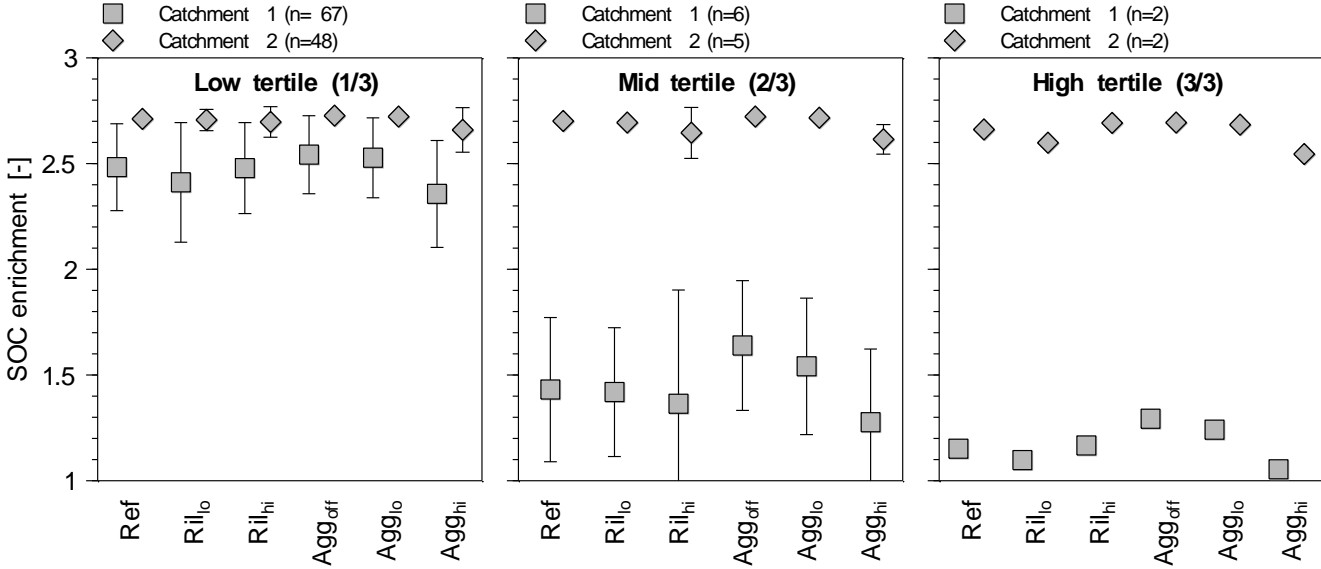

**Figure 6: Event size specific simulated mean SOC enrichment in delivered sediments of catchment C1 and C2. Error bars indicate one standard deviation. A, B and C represent the smallest, middle, and largest 33.3% of all event-specific sediment delivery masses. For details regarding different model runs and abbreviations see Table 2.**

