# Peer review of "Process-oriented modelling to identify main drivers of erosioninduced carbon fluxes"

_SOIL, 2016_

## Referee Comment (RC1) · Anonymous Referee #1 · 27 Dec 2016

Dear Editor and Authors,

The manuscript entitled "Process-oriented modeling to identify main drivers of erosion-induced carbon fluxes" has coupled a process-oriented multi-class sediment transport model with a carbon turnover model, with the aim to identify the main driving factors of erosion-induced C fluxes in two different catchments. The manuscript is of good novelty, and insightfully points out the potential relevance of soil aggregation levels and inter-field capacity in catchment-scale soil–carbon redistribution. Apart from identifying the dominant effects of tillage erosion in vertical C fluxes, the authors also notified the potential influences of episodic erosion events in lateral C fluxes, especially in catchments of highly varying topographic features. From my side, there are only two major comments, and a couple of minor suggestions.

[Figure]

Major comments:

1. While the authors did a good job in isolating erosion-induced C fluxes from the entire vertical C fluxes, all the fluxes were mainly based on erosional and depositional sites through dynamic replacements via monitoring yields and manure inputs (excuse me if I misunderstood you). The potential C fluxes during transport could arguably also play a role in vertical C fluxes, especially when taking aggregation levels into account. The de-/re-aggregation processes occurring during transport, and the accordingly induced mineralization and encapsulation, could potentially skew the overall C balances in individual catchments (see literature: Billings et al., 2010; Hu et al., 2016; etc.). To better justify the implication of this study, it would be nice to add some discussions on the possible impacts of mineralization during transport

2. The tillage component in the modeling set of this study is mainly adapted from a model derived from topography and tool-specific tillage erosion coefficients. Just out of curiosity, does it take into account for seasonal coincidence of tillage practices and rainfall frequency? For instance, is it possible that the tillage/water erosion effects to be enhanced when bare soil after tillage practices in spring receives frequent rainfall events (no need to be heavy)? Will these coupling effects have impacts on lateral and vertical C fluxes? Is it possible to be accounted for in current or future modeling settings?

Minor comments:

1. Just a personal suggestion, not a request, it may be better to some include some keywords in the manuscript title to reflect your key findings.

2. Better to have consistent terms when coming to "inter-field" or "intra-field" connectivity. 3. Page 10, from L7 to L11: it is just too long a sentence. Better to split into two.

Literature:

Billings SA, Buddemeier RW, de Richter BD, van Oost K, Bohling G (2010) A simple method for estimating the influence of eroding soil profiles on atmospheric CO2. Glob Biogeochem Cycles 24:1–14. doi:10.1029/2009GB 003560 Cambardella

Hu Y., Berhe A.A., Fogel M.L., Heckrath G.J., Kuhn N.J. (2016) Transport-distance specific SOC distribution: Does it skew erosion induced C fluxes? Biogeochemistry, 128 (3), 339-351: DOI 10.1007/s10533-016-0211-y

---

## Referee Comment (RC2) · Anonymous Referee #2 · 4 Feb 2017

This study represents a novel and needed step in the modeling of soil carbon redistribution and dynamics as a function of soil erosion and redistribution. It is well written and well documented and assumptions and limitations are clearly communicated. It is unfortunate that a decades long data set is not available to validate the model, but to the best of my knowledge, such a data set is not currently available. This does point to the need for funding for long-term monitoring at selected sites that would, eventually, provide the types of data needed to provide true validation of this and similar models. It may be worth providing a call for such monitoring in the conclusions to this study.

Page 1, Line 20 – I suggest "parent soil" rather than "parent material". Parent material has a very specific connotation in soil science, and it does not refer to soil materials that have been eroded and transported.

Page 3, Line 7 – do not capitalize loess.

Page 3, Lines 9-10 – What is the average depth of the Ap horizons?

Page 3, Line 27 – Put "(ICBM)" behind "Introductory Carbon Balance Model" and before the references.

Page 5, Line 15 – I believe the authors meant "rigorous validation" here, not "rigid validation"

Page 6, Line 9 – More detail on the soil survey should be given. Was this a detailed soil survey conducted expressly in support of research conducted in the study fields, or was the raster developed from a more general survey? What was the scale of the mapping? Is this survey readily available somewhere for readers to review? If so, please provide the reference.

Page 6, Line 17 – I would refer to these as "loess-derived" soils, not "loess-burden" soils. Same comment on Page 7, Line 8.

---

## Author Comment (AC1) · 22 Feb 2017

**RC1 Point-by-point response**
**Process-oriented modelling to identify main drivers of erosion-induced carbon fluxes**

Florian Wilken, Michael Sommer, Kristof Van Oost, Oliver Bens, Peter Fiener

*We are happy that the referee acknowledges the relevance of the study and support the publication. However, we carefully revised the manuscript according to the given recommendations. The referee misses explanation about the potential effect of mineralization during transport and the seasonal distribution of soil conditions due to tillage operations. We gratefully thank the reviewer for the careful advices and changed the manuscript accordingly. Please see the detailed answers (in italics) to the comments below:*

Major comments

While the authors did a good job in isolating erosion-induced C fluxes from the entire vertical C fluxes, all the fluxes were mainly based on erosional and depositional sites through dynamic replacements via monitoring yields and manure inputs (excuse me if I misunderstood you). The potential C fluxes during transport could arguably also play a role in vertical C fluxes, especially when taking aggregation levels into account. The de-/re-aggregation processes occurring during transport, and the accordingly induced mineralization and encapsulation, could potentially skew the overall C balances in individual catchments (see literature: Billings et al., 2010; Hu et al., 2016; etc.). To better justify the implication of this study, it would be nice to add some discussions on the possible impacts of mineralization during transport.

*We are well aware of the discussion regarding potential C mineralisation during transport. However, for the small catchments transport as such will only take place for a few hours and during this time C mineralisation via microbes is limited due to water saturated conditions. Hence, we do not expect any substantial C mineralisation during the short transport time. However, there might be an additional mineralisation following erosion, transport and deposition processes. Especially, at depositional sites there are some results from literature which imply a slight increase in SOC mineralisation of deposited material shortly after deposition (Hu et al. 2016; Van Hemelryck et al. 2010; 2011) but this additional mineralisation is small and governed by several side conditions (e.g. soil moisture of upper millimetre of soil; soil crusting and breakdown of crusts; transport of macro aggregates; Van Hemelryck et al. 201). Therefore, we did not include this in our modelling approach. However, we thank the reviewers for their important comment and we will added some information to the discussion section 4.1:*

*"The model also does not account for changes in C mineralization at depositional sites that may occur as a result of aggregate breakdown shortly after deposition (Hu et al., 2016; Van Hemelryck et al., 2010; 2011). However, the potential underestimation of C mineralisation at depositional sites is assumed to be small (< 2% at a loess site in Belgium; Van Hemelryck et al., 2011). In addition, various drivers of additional C mineralisation at depositional sites have been discussed in literature (soil moisture, crusting and crust recovery, deposition of large macro aggregates; Van Hemelryck et al., 2010; 2011) but there is still a substantial lack in process understanding. At this moment, this issue makes it difficult to transfer the specific experimental results into a modelling framework addressing other environmental conditions."*

The tillage component in the modeling set of this study is mainly adapted from a model derived from topography and tool-specific tillage erosion coefficients. Just out of curiosity, does it take into account for seasonal coincidence of tillage practices and rainfall frequency? For instance, is it possible that the tillage/water erosion effects to be enhanced when bare soil after tillage practices in spring receives frequent rainfall events (no need to be heavy)? Will these coupling effects have impacts on lateral and vertical C fluxes? Is it possible to be accounted for in current or future modeling settings?

*Tillage operations widely influence simulated water erosion due to changes in soil cover, crusting, roughness and manning's n. Hence, the tillage has a wide effect on lateral C fluxes, but the vertical C fluxes are calculated annually.*

*An important process which is not included is that the model does not account for tillage-induced changes in the soil physical properties that may influence the erodibility and infiltration of the soil itself. Fiener et al. (2013) showed that the time since tillage is one of the controlling parameters for the infiltration. The referee raised a good point and the following sentence is included to the section 4.2:*

*"Static soil parameters are to some extent an oversimplification and ignore feedback mechanisms that might be considered in future modelling studies of coupled water and tillage erosion (e.g. soil stability due to disruption, infiltration capacity; Fiener et al., 2013)."*

Minor Comments

Just a personal suggestion, not a request, it may be better to include some keywords in the manuscript title to reflect your key findings.

*Thank you for this hint! We extensively discussed this, but concluded that we would prefer to keep the title as short as it is.*

Better to have consistent terms when coming to "inter-field" or "intra-field" connectivity.

*Thank you! We mill consistently use the term inter-field connectivity.*

Page 10, from L7 to L11: it is just too long a sentence. Better to split into two.

*Indeed, this sentence is much too long. We split the sentence:*

*"Catchment C1 follows the expected behaviour, i.e. decreasing SOC enrichment with increasing event size (Auerswald and Weigand, 1999; Menzel, 1980; Polyakov and Lal, 2004; Sharpley, 1985), and is in good agreement with the results of Wang et al. (2010) for similar soils in the Belgian loam belt. In contrast, event size had hardly any effect on the SOC enrichment in catchment C2, where any larger particles, including aggregates, are deposited in the grassed waterway due to consistently*

5 *high hydraulic roughness throughout the year."*

References

Auerswald, K. and Weigand, S.: Eintrag und Freisetzung von P durch Erosionsmaterial in Oberflächengewässern, VDLUFA-Schriftenreihe, 50, 37-54, 1999.

10 Fiener, P., Auerswald, K., Winter, F., and Disse, M.: Statistical analysis and modelling of surface runoff from arable fields in central Europe, Hydrol. Earth Syst. Sci., 17, 4121-4132, 2013.

Hu, Y. X., Berhe, A. A., Fogel, M. L., Heckrath, G. J., and Kuhn, N. J.: Transport-distance specific SOC distribution: Does it skew erosion induced C fluxes?, Biogeochemistry, 128, 339-351, 2016.

Menzel, R. G.: Enrichment ratios for water quality modeling. In: CREAMS, Knisel, W. G. (Ed.), USDA Cons. Res. Rep.,
15 1980.

Polyakov, V. O. and Lal, R.: Soil erosion and carbon dynamics under simulated rainfall, Soil Sci., 169, 590-599, 2004.

Sharpley, A. N.: The selective erosion of plant nutrients in runoff, Soil Sci. Soc. Am. J., 49, 1527-1534, 1985.

Van Hemelryck, H., Fiener, P., Van Oost, K., Govers, G., and Merckx, R.: The effect of soil redistribution on soil organic carbon: an experimental study, Biogeosciences, 7, 3971-3986, 2010.

20 Van Hemelryck, H., Govers, G., Van Oost, K., and Merckx, R.: Evaluating the impact of soil redistribution on the in situ mineralization of soil organic carbon, Earth Surf. Processes Landforms, 36, 427-438, 2011.

Wang, Z., Govers, G., Steegen, A., Clymans, W., Van den Putte, A., Langhans, C., Merckx, R., and Van Oost, K.: Catchment-scale carbon redistribution and delivery by water erosion in an intensively cultivated area, Geomorphology, 124, 65-74, 2010.

---

## Author Comment (AC2) · 22 Feb 2017

**RC2 Point-by-point response**
**Process-oriented modelling to identify main drivers of erosion-induced carbon fluxes**

Florian Wilken, Michael Sommer, Kristof Van Oost, Oliver Bens, Peter Fiener

*We are happy that the referee acknowledges the relevance of the study and supports the publication. However, we revised the manuscript according to the given recommendations. The referee misses information about the soil survey and points at a number of wording errors. We gratefully thank the reviewer for the careful advices and changed the manuscript accordingly.*

*Please see the detailed answers (in italics) to the comments below:*

Page 1, Line 20 – I suggest "parent soil" rather than "parent material". Parent material has a very specific connotation in soil science, and it does not refer to soil materials that have been eroded and transported.
*Thank for pointing at this! We will use the term parent soil.*

Page 3, Lines 9-10 – What is the average depth of the Ap horizons?
*The average tillage depth is 0.2 m. See the sentence in section 2.1:*
*"For the established mulch tillage system, the main soil tillage operation was performed with a chisel cultivator (tillage depths approx. 0.2 m)."*

Page 3, Line 27 – Put "(ICBM)" behind "Introductory Carbon Balance Model" and before the references.
*Thank you! Will change it accordingly.*

Page 5, Line 15 – I believe the authors meant "rigorous validation" here, not "rigid validation"
Thank you for pointing at this wording error! We will use rigorous validation instead.

Page 6, Line 9 – More detail on the soil survey should be given. Was this a detailed soil survey conducted expressly in support of research conducted in the study fields, or was the raster developed from a more general survey? What was the scale of the mapping? Is this survey readily available somewhere for readers to review? If so, please provide the reference.
Thank you to point us at this missing reference. A detailed description of the soil survey can be found in Sinowski et al. (1997). We add the reference to the sentence in section 2.6:

*"(iii) soil data taken from a 50 x 50 m raster sampled during the soil survey in 1990/91 (Sinowski et al., 1997),"*

Page 6, Line 17 – I would refer to these as "loess-derived" soils, not "loess-burden" soils. Same comment on Page 7, Line 8. *We will use the term loess-derived. Thank you!*

5

Reference
Sinowski, W., Scheinost, A. C., and Auerswald, K.: Regionalization of soil water retention curves in a highly variable soilscape, II. Comparison of regionalization procedures using a pedotransfer function, Geoderma, 78, 145-159, 1997.